# Linking Urban Water Management, Wastewater Recycling, and Environmental Education: A Case Study on Engaging Youth in Sustainable Water Resource Management in a Public School in Casablanca City, Morocco

Hajar Nourredine [1,*], Matthias Barjenbruch [1], Angela Million [1], Btissam El Amrani [2], Nihad Chakri [2] and Fouad Amraoui [2]

1 Department of Urban Water Management, Technical University of Berlin, 13355 Berlin, Germany; angela.million@gmail.com (A.M.)
2 Faculty of Science Ain-Chock, University of Hassan II Casablanca, Casablanca 20000, Morocco
* Correspondence: hajar.nourredine@campus.tu-berlin.de

**Abstract:** The management of water resources is crucial for sustainable development, necessitating innovative solutions to address the increasing demand for safe water. Alternative approaches must be adopted to effectively engage young generations in understanding the importance of water resources. This chapter reports on an experiment that aimed to promote sustainability education by linking wastewater treatment and reuse with an educational garden. In particular, an undertaking was executed to establish a decentralized wastewater treatment system wherein purified water was employed for the purpose of irrigation. The study's primary focus is on the association between urban water management, wastewater recycling, and environmental education. The study has two distinct components. The first segment discusses three examples of projects that have employed urban wastewater treatment and reuse to generate environmental education materials using various approaches. The second component features a case study of a public high school in Casablanca, where students participated in a questionnaire and participatory workshops to design an educational garden. The study's outcomes include a proposed educational garden design that will be presented to the relevant authorities and project partners.

**Keywords:** environmental education materials; sustainability education; water resources management; wastewater treatment and reuse

## 1. Introduction

Water is one of the world's most critical resources, and its management is vital for sustainable development [1]. With the increasing demand for safe water, innovative solutions are necessary to ensure its availability for future generations. Alternative approaches must be adopted to engage young people in understanding the importance of water resources [2]. One such approach is to link wastewater treatment and reuse with education, promoting sustainability education [3].

Integrating education for sustainable development (ESD) into the education systems of most countries in the Global South is a significant challenge, especially considering the pressing need to address environmental degradation [4]. Morocco is a case in point, where the costs of environmental degradation are estimated to be around 13 billion dirhams, equivalent to 3.7% of the country's GDP [5]. In recognition of the gravity of the situation, the country has embarked on an ambitious environmental transition. Environmental education has emerged as a top priority in this transition, as evidenced by the country's concerted efforts to address ESD issues [6].

Previous research has shown that environmental education is vital for promoting sustainable development [7]. The United Nations Educational, Scientific and Cultural

Organization (UNESCO) has recognized this and included education for sustainable development (ESD) as a key component of its Sustainable Development Goals (Berlin Declaration for 2030). Indeed, environmental engineering provides an interesting laboratory for learning about sustainable development in real life [8]. One laboratory can be wastewater management. Wastewater management is an interesting laboratory for learning about sustainable development because it involves several key sustainability principles [9]. First, it involves the conservation of resources, as treated wastewater can be reused for various purposes, such as irrigation or industrial processes [10]. Second, it involves pollution control, as wastewater treatment processes remove harmful pollutants from the water before it is discharged into the environment. Third, it involves public health, as untreated wastewater can pose significant health risks to communities [11].

According to Million et al. [12], wastewater management provides an excellent opportunity for students to learn about sustainable development principles in a real-world context. By engaging in wastewater management projects, students can learn about the technical, economic, and social factors that impact sustainability. For example, students can learn about the various treatment technologies available and the factors that influence their selection. They can also learn about the economic costs and benefits of different treatment options, as well as the social and cultural factors that impact public perceptions of wastewater management [13].

Research around wastewater management in the Global South has historically overlooked the potential for creating educational tools tailored specifically for children [14–16]. These projects have typically focused on technical aspects of wastewater management, such as treatment methods and infrastructure development, while neglecting the educational dimension that could engage and empower younger generations.

Johnson et al. [14] conducted a study on wastewater management initiatives in rural communities in India, highlighting the lack of educational resources targeting children. Similarly, Chen et al. [15] examined urban wastewater management projects in China, noting a scarcity of educational tools tailored to the younger population. López et al. [16] investigated wastewater treatment and reuse as an educational tool in a peri-urban community in Mexico, emphasizing the need to address this educational gap.

The oversight of incorporating children as a target audience in wastewater management research limits the potential for fostering environmental awareness, sustainable practices, and community engagement from an early age. By neglecting the educational aspect, these projects miss opportunities to develop a well-rounded understanding of wastewater management among future generations and hinder the potential for long-term sustainable practices.

Recognizing the significance of education for sustainable development, it is crucial to bridge this gap by designing research projects that explicitly consider the creation of learning tools for children. Incorporating interactive and age-appropriate educational materials, such as games, workshops, and curricula, can facilitate the transmission of knowledge and values related to wastewater management [17,18].

To address this gap, this paper presents fieldwork conducted with a public high school in Casablanca to explore how treated wastewater reuse can be used as an environmental education tool. The study aims to answer the following research questions: How can treated wastewater reuse be used as an environmental education tool in public schools? How has environmental education been integrated into the public school in Casablanca? The answer to the first question will be developed in two parts. The first part presents existing projects where wastewater treatment and reuse have been the basis for producing a learning tool for environmental education. The second part concerns the case study conducted in the public high school in Casablanca. Overall, the paper highlights the potential for wastewater management to serve as an effective learning tool for promoting sustainable development and offers practical insights for integrating environmental education into the curriculum.

The methodology used in this study involved implementing a decentralized wastewater treatment project and using the treated water for irrigation purposes in an educational

garden. The study used questionnaires and participatory workshops to engage students in designing the garden. The study's goal was to promote sustainability education, and the questions addressed the relationship between wastewater treatment, reuse, and environmental education.

## 2. ESD in Wastewater Management—Overview of Projects Combining Wastewater Management and Environmental Education

The concept of education for sustainable development (ESD) encompasses a wide range of academic disciplines. ESD aims to develop a set of key competencies, namely interdisciplinary knowledge, independent action, and active participation in social decision-making processes [19]. These goals are aligned with the Sustainable Development Goals (SDGs) established in 2015, which state that by 2030, all learners should possess the knowledge and skills necessary to promote sustainable development. This includes education on sustainable lifestyles, human rights, gender equality, a culture of peace and non-violence, global citizenship, and an appreciation of cultural diversity and its contribution to sustainable development [20]. In this regard, ESD initiatives such as Water Fun for Life!, Roof Water Farm in Berlin, and the School-Garden in Morocco UAC project, serve to prioritize sustainable development education. During our survey, it was very important to look for projects where wastewater treatment and/or its reuse was used to promote environmental education. In the following section, we will present and analyze those projects that have successfully combined both fields.

Ecological wastewater treatment technologies have emerged as a sustainable alternative to centralized wastewater treatment systems, particularly in contexts where connection to a central wastewater treatment plant is not a cost-effective solution, such as in small rural settlements located far from a central wastewater treatment plant. Decentralized wastewater treatment systems based on ecological wastewater treatment technologies provide an effective approach to treating wastewater while reducing the negative impact on the environment. However, despite their potential benefits, the adoption of these technologies faces several challenges, including a lack of awareness and knowledge of wastewater treatment among the population.

These projects include Water Fun for Life!, Roof Water Farm in Berlin, and the UAC School-Garden project in Morocco, all of which demonstrate a commitment to prioritizing education for sustainable development. While these projects encompass diverse approaches and operate within different contexts, they share similarities with our practical case study. By drawing upon the findings and insights derived from these projects, we were able to develop the working methodology for our case study.

Water Fun for Life! is an exemplary project that focuses on wastewater treatment and reuse as a means of educating communities about water conservation and environmental sustainability [21]. Similarly, the Roof Water Farm project in Berlin utilizes innovative technologies to treat and reuse wastewater while simultaneously providing educational opportunities to raise awareness about sustainable water management [22]. Additionally, the UAC School-Garden project in Morocco integrates wastewater treatment and reuse into the school curriculum, promoting environmental education and sustainable practices among students [23].

The selection of these three projects was based on their relevance to our case study, as they provided valuable insights and methodologies that informed our research approach. Furthermore, the availability and accessibility of data were considered in the selection process to ensure that our study would be robust and comprehensive.

By integrating the knowledge gained from these projects into our research, we aim to enhance the effectiveness and applicability of our case study. The insights derived from these projects, coupled with our empirical observations, will enable us to develop a robust working methodology that aligns with established best practices and advances our understanding of wastewater treatment and reuse as an educational tool for sustainable development.

*2.1. Case 1: Water Fun. . . for Life!*

One notable initiative developed within this framework is the educational program "Water Fun—Hands, Minds, and Hearts on the Water for Life!" This program seeks to provide comprehensive information on ecological wastewater treatment technologies and their design, operation, and maintenance. Additionally, the program aims to demonstrate the benefits of these technologies, including the reuse of treated wastewater for agricultural purposes. The program has been implemented in two distinct regions, namely the Lower Jordan Valley and the Kharaa River Basin. These regions were chosen due to their unique characteristics, including their location and the challenges they face in terms of wastewater management.

Information on the program and its implementation has been gathered from various sources, including the project coordinators and their website (http://www.waterfunforlife.de/, accessed on 1 November 2022). The program is aimed at fostering a culture of sustainability by promoting the adoption of ecological wastewater treatment technologies and reducing the negative impact of wastewater on the environment. By linking wastewater treatment and environmental education(Figure 1), the program seeks to better understand the importance of sustainable wastewater management and promote the adoption of ecological wastewater treatment technologies as a viable solution to wastewater management challenges in decentralized settings.

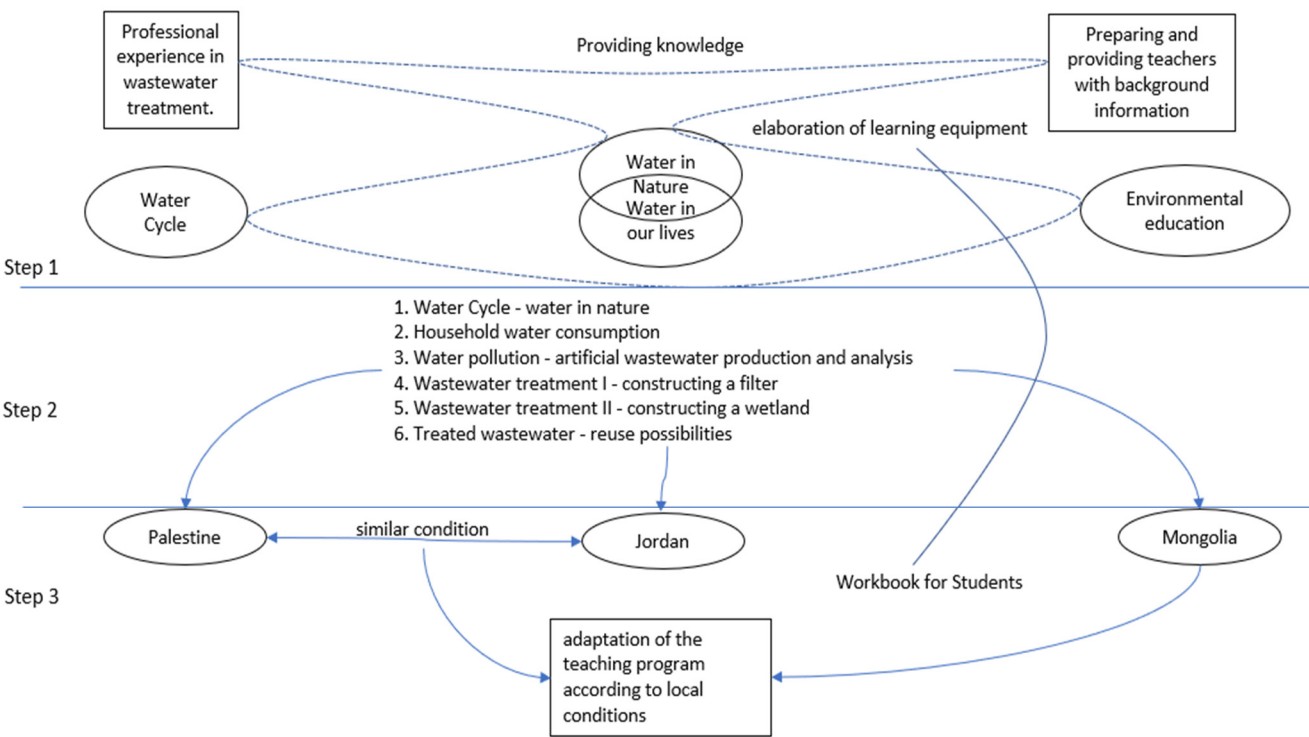

**Figure 1.** The process of introducing environmental education in the framework of the Water Fun. . . For Life project [24].

The teaching program units are designed in collaboration with local partners to address the particular challenges related to water resources and management in Jordan, Palestine, and Mongolia. The primary objective of the program "Water Fun. . . For Life!" is to facilitate critical thinking and reflection among students and teachers alike on topics such as water consumption, wastewater generation and composition, wastewater treatment, and the potential for treated wastewater reuse [25]. By engaging participants in discussions and activities surrounding these topics, the program encourages individuals to develop a more discerning understanding of knowledge construction and co-construction in the context of water management [26]. This critical approach empowers participants to question existing

assumptions, challenge traditional notions, and explore alternative perspectives to promote a more comprehensive understanding of water-related issues [27]. Through this program, students and teachers alike have the opportunity to critically engage with the complexities of water management and contribute to the generation of innovative solutions.

The program is intended to introduce participants to the benefits and opportunities associated with decentralized wastewater treatment systems for irrigation in Jordan and Palestine, as well as for willow production in Mongolia. To this end, the teaching program units (Table 1) incorporate a range of classroom activities and experiments aimed at fostering a deeper understanding and appreciation of the importance of treated wastewater as a valuable resource. Through hands-on learning experiences, students will be equipped with the knowledge and skills necessary to make informed decisions about water use and management in their respective communities. The ultimate goal of the program is to empower students to become active agents of change, promoting sustainable water use practices and contributing to the long-term conservation of water resources in their respective regions.

**Table 1.** Teaching program units [24].

| Teaching programs in Jordan & Palestine | | Teaching program in Magnolia |
|---|---|---|
| **Unit 1: Water cycle—water in nature** Students learn about the water cycle and its processes, the different states of water, and how it affects water quality. | | |
| **Unit 2: Household water consumption—water in our life** This unit introduces students to the topic of household water consumption so that they can identify where they use water at home in their everyday activities and how wastewater is generated through this use. They get to know where the wastewater goes when it leaves their homes and learn about the need to treat wastewater to avoid environmental pollution and health problems. | | |
| **Unit 3: Water pollution—artificial wastewater production and analysis** Students produce artificial wastewater with safe ingredients and observe how the water quality changes. They learn the principles of water analysis by measuring simple parameters. | | |
| **Unit 4: Wastewater treatment I—constructing a filter** Students experiment with various materials to construct a bottle filter as a simple treatment system for the wastewater generated in the previous unit. By comparing the results from the different filters, students learn that each material has a specific functionality in the filtration process. | | |
| **Unit 5: Wastewater treatment II—Constructing a wetland** Students study wetlands on a model scale as a modern ecological technology that "copies" natural mechanisms for wastewater treatment. They demonstrate the functionality of this technology by subsequent water analysis: they measure the quality of the treated wastewater to decide if it can be reused or not according to a country-specific water standard. | **Unit 5: Wastewater treatment II—improving the filter** In this unit, students reflect on the best combination of materials to remove more pollutants from the artificial wastewater and construct improved filters. Students measure the quality of the filtered wastewater to decide whether it can be reused according to a specific water standard. | |
| **Unit 6: Treated wastewater—reuse possibilities** Students discharge the treated wastewater with quality lower than the standard into the toilet. Students learn that treated wastewater with the same or higher quality than the standard can be used for the irrigation of crops and that treated wastewater is a valuable resource. | | |
| **Unit 7: Excursion to a wastewater treatment plant** | | |
| Teaching and learning outside the classroom as part of environmental education for sustainable development in Jordan and Palestine. | Teaching and learning outside the classroom as part of environmental education for sustainable development in Darkhan, Mongolia | |

### 2.2. Case 2: Roof Water Farm in Berlin

The Roof Water Farm project in Berlin serves as an innovative model that integrates wastewater treatment technology with food production in a "closed-loop urban agriculture approach". The project incorporates hydroponics and aquaponics as building-integrated,

water-based agricultural strategies [28]. The successful implementation of the project has resulted in a comprehensive knowledge base on the integration of wastewater reuse and urban agriculture (Figure 2). The knowledge and experience gained from this project have been utilized to create training courses offered by the Roof Water Farm project to promote and transfer the knowledge to various communities. The project serves as a learning tool that creates awareness and promotes daily sustainable practices for society, neighborhood residents, and students through workshops [12]. Its significance is in promoting the knowledge transfer of sustainable urban agriculture practices.

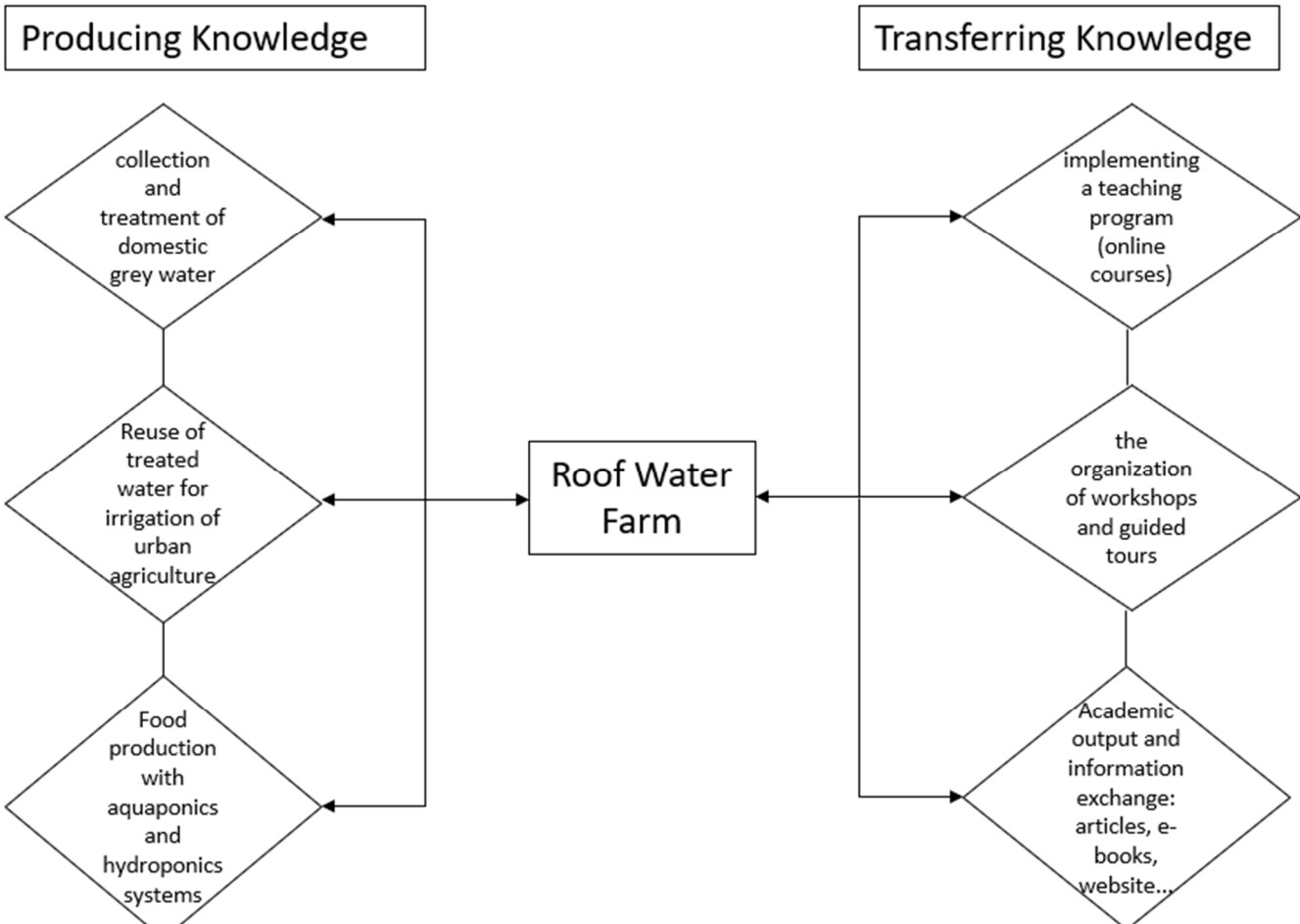

**Figure 2.** Generation and transfer of knowledge through the Roof Water Farm project [12].

*2.3. Case 3: School-Garden in Morocco UAC Project*

The Urban Agriculture Project (UAC), a Moroccan–German research initiative funded by the German Federal Ministry of Education and Research (BMBF), has recently initiated the development of an ecological hammam in the Casablanca region. The project aims to promote sustainable urban development through the implementation of urban agriculture practices, which involve formal or informal agricultural production within a city. One of the UAC's initiatives, the "solitary farm," is a community garden in Douar Ouled Ahmed, which provides local women with training in organic farming practices and enables them to generate income through the sale of agricultural products [29]. To promote energy efficiency and reduce the environmental impact of the project, water from the local hammam is treated in a constructed wetland and reused for irrigation purposes in the community garden. Furthermore, solar thermal collectors will be installed on the hammam's roof to replace the traditional wood-burning oven and heat the bath water [30]. Another initiative of the UAC is the implementation of a school garden managed by an elementary school, which aims to

educate students on the principles of organic farming and promote the themes of healthy food and sustainability. Figure 3 summarizes how the solidarity farm has been built up. This project intends to empower students to act as advocates of urban agriculture practices and environmental protection.

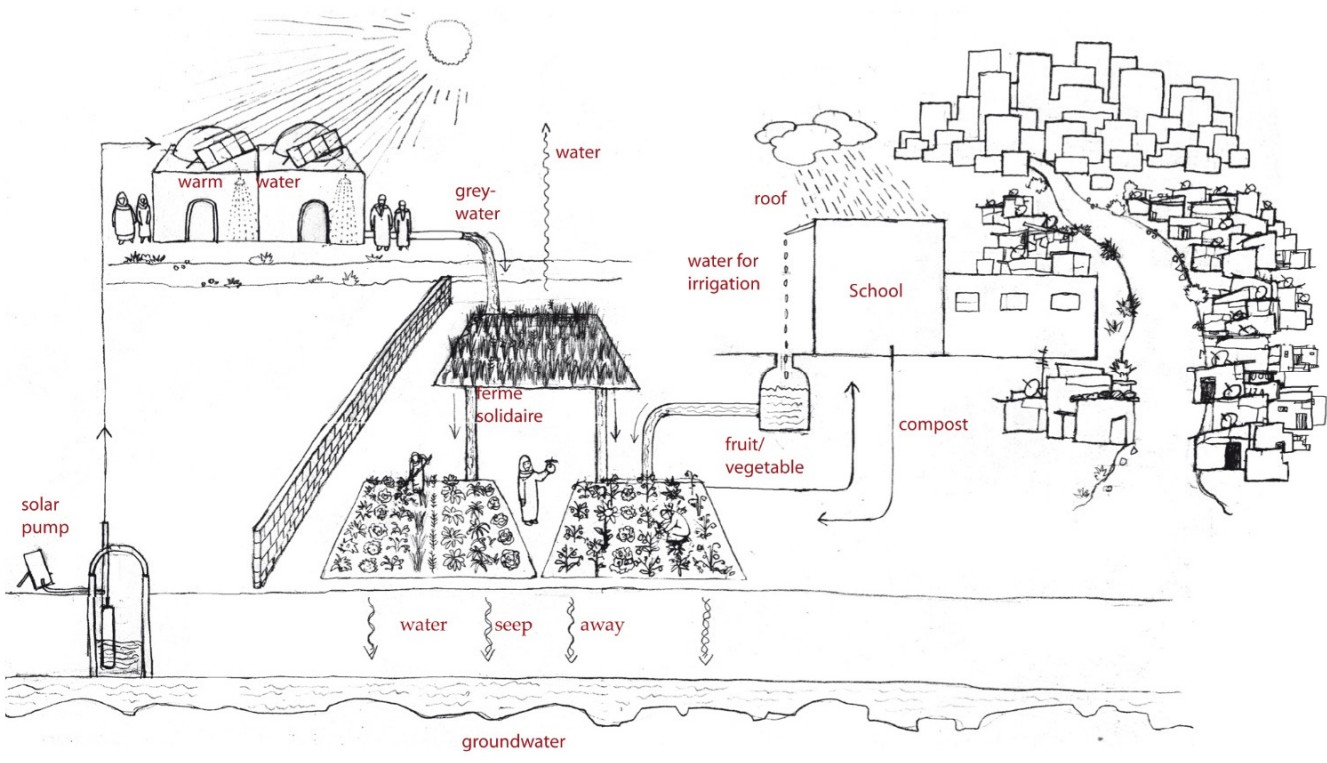

**Figure 3.** Informal settlement graphic of Pilot Project 2: Urban Agriculture [29].

Empowerment within this research manifests through multiple dimensions. Firstly, students are actively involved in the planning and implementation of urban agriculture practices, allowing them to take ownership of their learning and contribute to the development of sustainable solutions [31]. This participatory approach fosters a sense of agency and self-efficacy among students, empowering them to make a positive impact on their environment.

Furthermore, the project encourages students to engage in advocacy activities, raising awareness about the importance of environmental protection and sustainable practices within their communities [32]. By assuming the role of advocates, students develop leadership skills, enhance their communication abilities, and become catalysts for change.

Several researchers have been inspired by the UAC's concept and are continuing to work on retrofitting traditional hammams for environmentally friendly design. The overarching objective of the UAC project is to examine the relevance of urban agriculture in contributing to sustainable urban development [3].

*2.4. Insights Drawn from the Case Studies*

The three projects presented in this case study highlight the importance of sustainable water management and urban agriculture practices in promoting sustainable development in different regions. The teaching program units in Jordan, Palestine, and Mongolia aim to equip students with the knowledge and skills necessary to make informed decisions about water use and management in their respective communities, ultimately empowering them to become active agents of change in promoting sustainable water use practices. The Roof Water Farm project in Berlin serves as an innovative model that integrates wastewater treatment technology with food production, promoting daily sustainable practices for

society, neighborhood residents, and students through workshops. Finally, the UAC project in Morocco aims to promote sustainable urban development through the implementation of urban agriculture practices, such as the school garden and community garden, which provide local women and students with training in organic farming practices and promote the themes of healthy food and sustainability. The initiatives undertaken in these projects serve as a significant learning tool for promoting sustainable practices and environmental protection, ultimately contributing to the long-term conservation of natural resources in their respective regions.

## 3. Environmental Education Implementation at a High School in Casablanca

For almost six years, the LYDEC Foundation has been providing support to the Action and Research for Sustainable Development (ARADD) association to address sustainable development issues. The ARADD-URBAN concept has resulted in several successful projects, such as the Eco-Hammam project and the Urban Agriculture Pedagogical Garden of the WTP of Mediouna (which received the Hassan II Prize for the Environment in 2018). This concept has inspired many researchers and stakeholders to continue exploring sustainable hammam models while incorporating other objectives related to the United Nations Sustainable Development Goals (SDGs) in Morocco. This research project is part of the ARADD and the LYDEC Foundation partnership and is expected to last for three years. The project aims to create a demonstrative model that combines new technology for water treatment and reuse with energy savings.

The present study intends to install a wastewater treatment and reuse system in a public school to promote environmental education among students and staff. The selection of the school for this project aims to integrate environmental education into the daily lives of its members and to contribute to building a safe and sustainable environment for education through the educational garden.

The study area, Dar Bouazza, is a suburban municipality located in the Province of Nouaceur, in the Casablanca-Settat region of Morocco. The municipality is bordered by the Atlantic Ocean to the north, the municipality of Casablanca to the east, the rural municipality of Ouled Azzouz to the south, and the rural municipality of Soualem Trifiya of the province of Berrechid to the west. It is divided into five administrative annexes: the Ben Abid annex, the Dar Bouazza Centre annex, the Errahma I annex, the Errahma II annex, and the Ouled Ahmed annex. The population of Dar Bouazza is estimated at 151,373 inhabitants, with a growth rate of approximately 10% [33]. Additionally, the municipality is home to several large-scale economic and tourist housing projects, as well as shanty towns and douars.

Douar and shanty towns are two distinct types of settlements that differ in their characteristics and contexts. A douar typically refers to a small rural or peri-urban settlement commonly found in North Africa, particularly in countries such as Morocco, Algeria, and Tunisia. Douars are characterized by more permanent dwellings and are often situated in rural areas [33]. On the other hand, shanty towns are urban areas associated with substandard living conditions and makeshift structures.

The urban development in the area has led to an increase in public facilities, such as schools, mosques, ovens, and hammams, to meet the needs of both regulated and unregulated (random) populations [33].

The study was conducted in the public high school Haj Ahmed Naciri, located in the administrative annex Errahma II in Dar Bouazza (Figure 4). The high school was selected due to its location, which is opposite a public hammam, and its high enrollment of approximately 1300 students, making it the largest high school in the municipality. Additionally, the school was identified as requiring green spaces, and the administrative staff was committed and motivated to engage in environmental education initiatives.

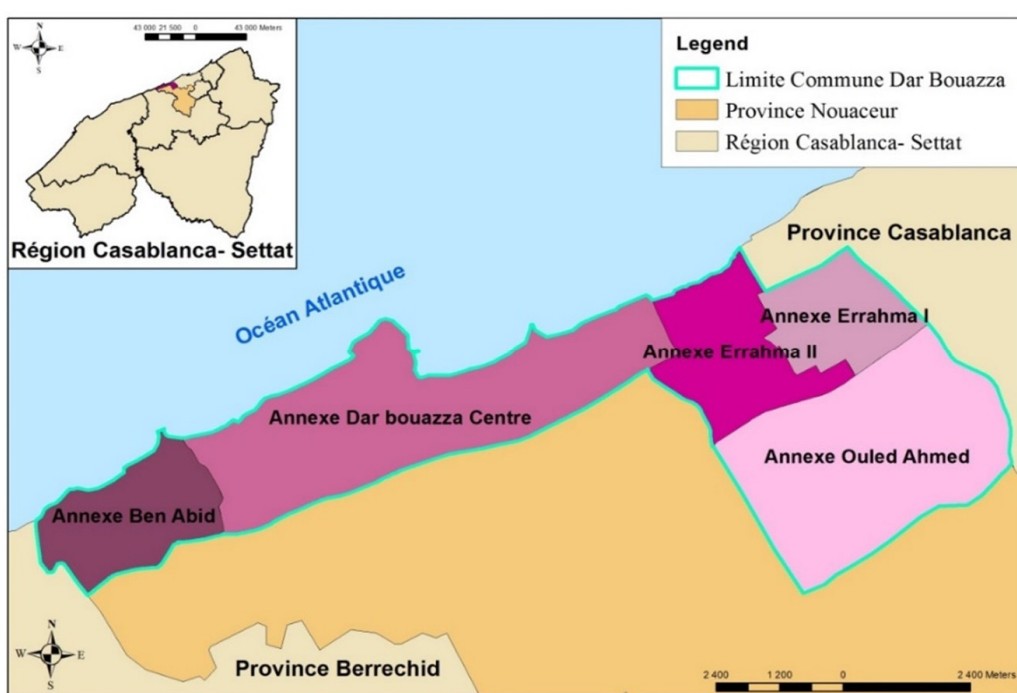

**Figure 4.** Map of the study site (Municipality Dar Bouazza) [33].

The school premises covers an area of 4436 m². However, despite the large area, some surfaces are not very well managed. The space is mainly occupied by the sports field, the courtyard, the classrooms, and the administration building. This study aimed to address this issue by proposing the installation of a wastewater treatment and reuse system in the public hammam, which is located adjacent to the school. The proposed system is intended to enhance environmental awareness among students and staff by demonstrating the importance of conserving natural resources and promoting sustainability in their daily lives.

The high school Haj Ahmed Naciri is one of the major public schools in Dar Bouazza. The school is equipped with various facilities, including classrooms, laboratories, and a library. The school also has a sports field, which is used for various sporting activities. However, despite its various facilities, the school premises lack green spaces, which are important for promoting environmental education and contributing to the well-being of students and staff. Therefore, the proposed wastewater treatment and reuse system is expected to not only provide a practical solution for water management but also to create an opportunity to develop an educational garden that will be used to promote environmental education and sustainability.

Moreover, the choice of the school's location is strategic for the proposed project. The public hammam located opposite the school is a significant source of wastewater. The installation of the proposed wastewater treatment and reuse system in the hammam will provide a practical solution for water management, which can be used to irrigate the educational garden on the school premises. By using treated wastewater, the project will help to conserve scarce water resources in the region and promote sustainable water management practices.

In summary, the high school Haj Ahmed Naciri was selected as the site for the proposed wastewater treatment and reuse system due to its location, enrollment, and administrative staff's commitment to environmental education initiatives. The installation of the proposed system will address the lack of green spaces on the school premises, provide a practical solution for water management, and create an opportunity to develop an educational garden for promoting environmental education and sustainability. Furthermore, the proposed system will help conserve scarce water resources in the region and promote sustainable water management practices.

The case study was chosen through a collaborative effort involving the university, local stakeholders, institutions, and the community. The goal is to create a practical learning tool for public school students, focusing on wastewater treatment and reuse. The selection was based on the school's proximity to the public hammam, its available space, and its strong commitment to collaboration. These factors will enhance the study's relevance and provide valuable insights into real-world scenarios, benefiting both academic research and public awareness of sustainable wastewater management practices.

Figure 5 illustrates the school environment, reflecting the surroundings experienced by the students and influencing their daily lives. Notably, the presence of a hammam is depicted, serving not only as a traditional space but also as a vital hygiene resource for individuals who lack access to showers or hot water. The diverse housing structures featured in the image indicate that the public school accommodates students from various socio-economic backgrounds, each with distinct means and levels of sanitation infrastructure incorporated into their everyday routines.

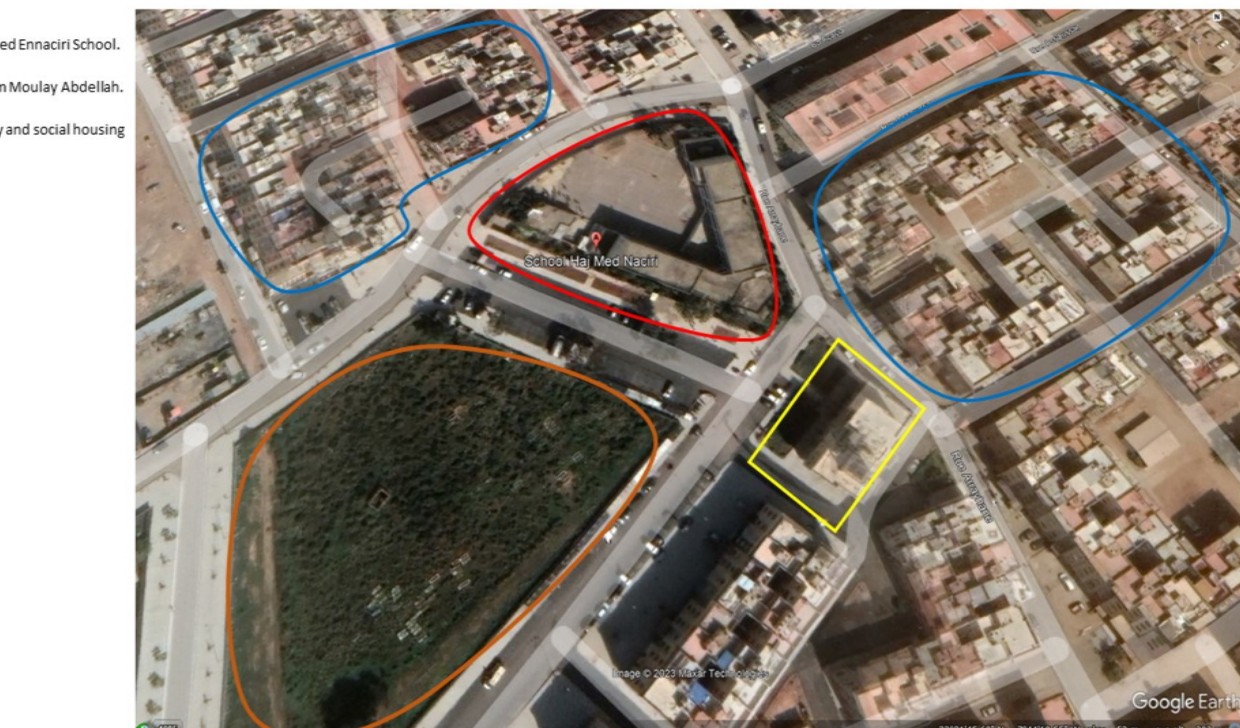

**Figure 5.** Location and description of the neighborhood of Haj Ahmed Naciri High School (Google Earth Pro).

*3.1. Methodology*

The methodology employed in this chapter is action research, which is recognized as an appropriate approach for achieving effective outcomes through collaboration with various stakeholders, including local actors [34]. Action research, as defined by Kalungwizi [34], is a research approach that emphasizes the active participation of stakeholders in the research process, encouraging them to collaboratively identify problems, propose solutions, and implement actions.

Action research is particularly valuable in the context of high school-level education, as it promotes alternative learning methods that emphasize active engagement and participation. This approach allows stakeholders to be involved in the planning, action, and reflection cycle, ensuring the sustainability of the learning program [35].

Numerous studies have underscored the efficacy of action research in projects that involve diverse stakeholders [34]. The decision to employ an action research approach for this study is driven by the active involvement of multiple stakeholders, encompassing the

school community (students, teachers, and administrative team), Eco-Hammam project managers, local associations, institutions, and public authorities. The rationale behind selecting this approach lies in its capacity to provide a participatory and inclusive framework that integrates all these stakeholders, ensuring their active participation in the project and promoting sustainability.

Moreover, in his study, Wilson [35] presented how action research enables meaningful engagement with stakeholders, resulting in improved project outcomes. Given the context of public schools in Morocco, where engaging students as stakeholders can be challenging, adopting an action research approach holds immense promise. By involving students in the environmental education process, this initiative can pave the way for a transformative shift in how such activities are perceived and implemented in public schools.

The project's objective was to establish a pedagogical garden within the environmental education (EE) framework at a public high school. The selection of the project location was strategic, with the school situated near a traditional hammam that served as the water source for the garden's irrigation system. The initial step involved presenting the project to the administrators of Haj Ahmed Naciri High School, emphasizing the importance of collaborative participation to encourage institutional ownership and support for the garden initiative.

Subsequently, the project was introduced to the staff and students using a questioning method (whether interviews with teachers or student questionnaires), which allowed for the expression of opinions and the assessment of motivation levels among participants [36]. This approach not only explained the purpose of the project but also fostered a sense of involvement and ownership among the stakeholders.

Collaboration between teachers and regional authorities was integral in the development of the questionnaire, ensuring its content and relevance to the study's objectives [37]. Following this collaboration, the questionnaire was distributed to the school's student body. The initial sample size comprised 1300 students, all of whom received the questionnaire.

Participation in answering the questionnaire was a requirement for students to engage in upcoming activities related to the study. The teachers emphasized this connection between questionnaire completion and participation in subsequent events. Students were informed that their responses would remain anonymous, with only age and sex information collected to maintain confidentiality and comply with ethical considerations [38].

The questionnaire was structured into three distinct sections. The first section aimed to assess the students' environmental knowledge and awareness level. The second section sought to gauge the extent to which students applied environmentally friendly practices in their daily lives. Lastly, the third section aimed to measure the students' motivation and commitment towards participating in the construction of the educational garden and attending workshops.

In the third phase of the study, two interactive workshops were organized specifically for the students of Haj Ahmed Naciri High School. The first workshop focused on wastewater treatment and reuse, providing students with valuable knowledge and insights into sustainable water management practices. The second workshop centered around landscape mapping within the school premises, enabling students to actively engage in the design and planning process of the garden. Students were divided into three groups, with each group supervised by a teacher from the high school. The aim was to encourage inclusive participation and empower students to contribute to the realization of their vision for the garden based on their daily needs and preferences [39]. Importantly, these workshops were instrumental in fostering inclusive student participation through an action-reflection cycle. This approach enabled students to actively engage in hands-on activities, reflect on their experiences, and refine their ideas and designs. By incorporating this iterative process, the workshops promoted critical thinking, creativity, and active involvement of the students in shaping their own learning experiences.

These workshops served as pivotal moments for collaborative learning, as students actively participated in the action-reflection cycle. By engaging in hands-on activities and

reflecting on their experiences, students were able to develop a deeper understanding of environmental issues and sustainable practices. Moreover, the involvement of teachers from the high school ensured the integration of the project into the curriculum and provided support and guidance throughout the process.

Through the utilization of action research methodology and the implementation of interactive workshops, this project aimed to enhance environmental education and promote sustainable practices among high school students. By fostering inclusive participation and empowering students to become active agents of change, the pedagogical garden initiative sought to cultivate a sense of environmental responsibility and create a lasting impact within the school community. A schematic diagram summarizing the order of the steps is presented in Figure 6. The results of our study are presented in the following section.

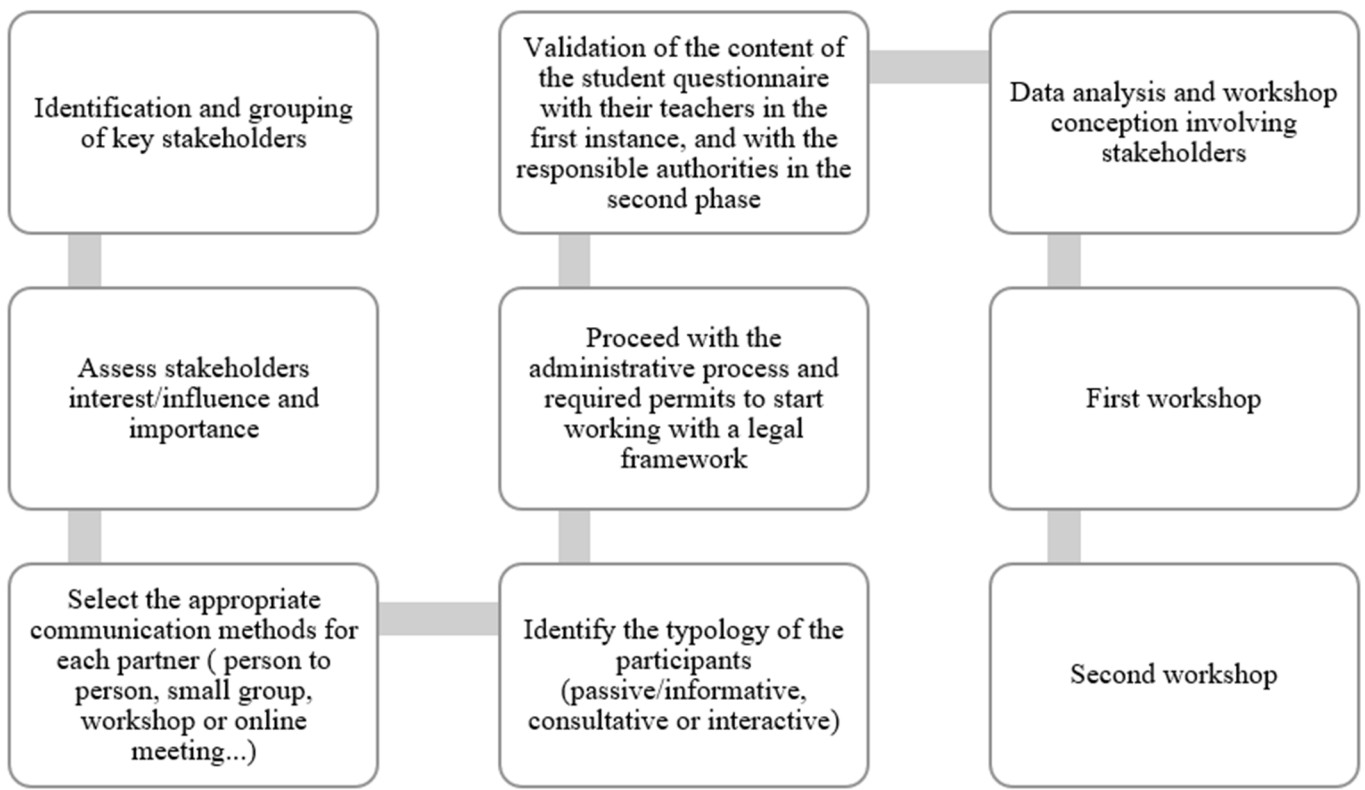

**Figure 6.** Methodology process adopted to work with the public school Haj Ahmed Naciri and introduce environmental education.

### 3.2. Results and Discussion

3.2.1. Outcome and Evaluation of the Student Questionnaire

Table 2 provides an overview of the survey respondents, which consisted of 33 students out of a total student population of 1300. The gender distribution among respondents was 44.1% male and 55.9% female, with the age range of participants falling between 15 and 18 years old. It is important to note that these results are not representative of the entire student population, as only those who chose to participate in the project were included in the sample. Further analysis and interpretation of the results are necessary to gain a more comprehensive understanding of the perceptions and attitudes of the student body towards the implementation of a pedagogical garden in their school.

**Table 2.** Characteristics of the survey respondents.

| Total Students in School | Number of Students Who Responded to the Questionnaire | Gender Percentage among Respondents | Age Range |
|---|---|---|---|
| 1300 | 33 Students | 44.1% Male 55.9% Female | 15–18 years old |

The questionnaire developed in collaboration with teachers was divided into three sections. The first section focused on gathering information about the responders' profiles, including their age, gender, and other relevant demographic information, as summarized in Table 2. The second section of the questionnaire was designed to assess the level of knowledge of environmental issues among the students and their understanding of the importance of environmental protection. This section aimed to gain insight into the student's level of knowledge and awareness of environmental issues so that it could be considered when designing the workshops. Additionally, this section aimed to identify the degree of commitment of respondents to environmental protection within their daily lives.

Table 3 and Figure 7 present the results of the third section of the questionnaire, which asked students to name at least two words they associated with the word "environment". The responses were then grouped into six categories based on their definitions, and each category was assigned a number to allow for correlations between the responses. The most frequently mentioned categories were water and flora, which are related to nature and public health. The number of responses in each category is provided in Table 3, with water being the most commonly cited category with 20 responses, followed by flora with 16 responses. The findings of this study suggest that the students possess a basic understanding of environmental issues, with a focus on nature and public health.

**Table 3.** The categories of words associated with the word "environment" and their associated numbers and word frequencies.

| Name at Least Two Words You Associate with "Environment". | Total |
|---|---|
| 1 = Nature | 10 |
| 2 = Public health | 14 |
| 3 = Fauna | 3 |
| 4 = Flora | 16 |
| 5 = Water | 20 |
| 6 = Gardens and farmland | 12 |
| **Total** | **75** |

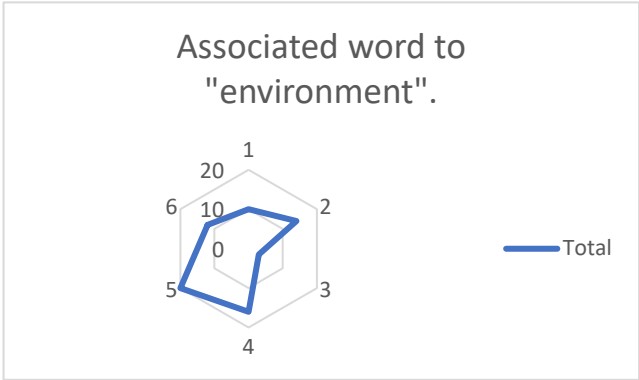

**Figure 7.** Correlation of the categories of words.

As shown in Figure 8, the questionnaire also aimed to identify the environmental issues that students perceived as having the most significant impact on their lives. The results indicate that water scarcity and desertification are the top environmental issues that students are concerned about, with 34% and 25% of respondents selecting them as the most pressing issues, respectively. These findings are in line with the challenges that Morocco is currently facing, as the country is among the most water-scarce countries in the world and climate change is expected to have severe consequences on the country's economy, society, and environment [40]. The identification of these environmental issues by high school students highlights the need for increased awareness and action to address these challenges, as the youth play a crucial role in shaping the future of their communities and the world.

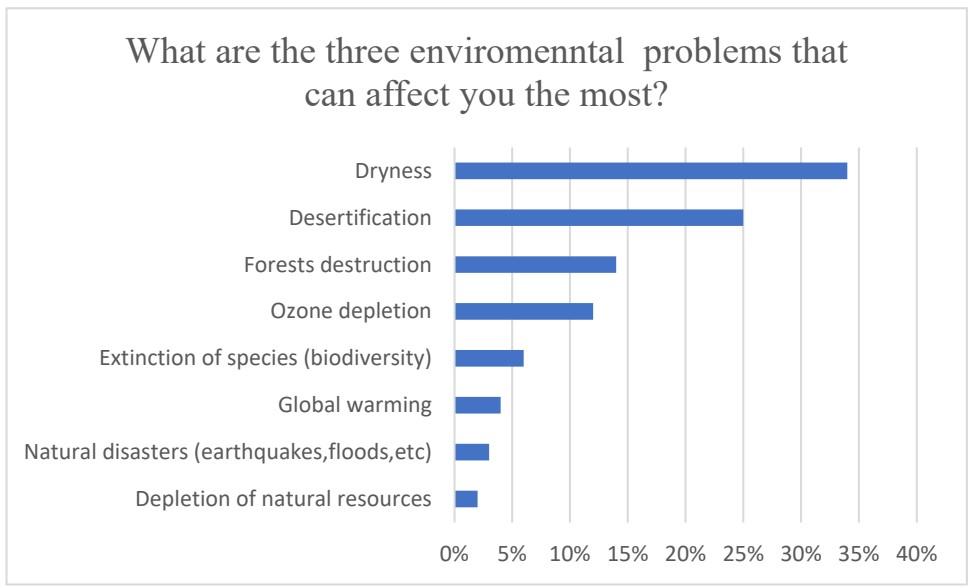

**Figure 8.** Ranking of environmental issues that may affect students.

The third section of the questionnaire aimed to assess the degree of involvement of the students in environmentally friendly actions in their daily life, as well as in their schools. Table 4 showed that the majority of students had simple daily actions to protect the environment, such as switching off electrical appliances, avoiding leaving the water running for no reason and throwing waste in the garbage containers. In terms of future actions, most students thought that they would engage in waste-selective sorting, save water, and save energy. Moreover, the majority of students (69.7%) indicated that they would engage in environmentally friendly actions in their neighborhoods in the future.

Regarding school activities related to the environment, the majority of students (60%) chose environmental education over other activities such as music and painting. Interestingly, 58.8% of the students were willing to engage in the activities of the environmental club that was created by the teachers in previous years. This indicates that students are interested in environmental activities and are willing to engage in such activities if they are provided with the necessary resources and support.

Overall, the results suggest that high school students in Morocco are aware of environmental issues and are willing to engage in environmentally friendly actions in their daily life and school activities. However, the study also highlights the need for schools to provide the necessary resources and support to facilitate the engagement of students in environmental activities. The findings of our study underscore the importance of schools in facilitating student engagement in environmental activities. While it is significant that students express their intentions to take certain actions, it is essential to recognize that intentions alone may not be sufficient to drive meaningful change. This aligns with the ex-

isting literature, which highlights the need to go beyond individual intentions and address structural and social barriers to promote behavior change [41].

**Table 4.** Students' answers to the questions in the questionnaire.

| Do You Have Simple Daily Actions to Protect the Environment? | Yes Always | Sometimes Yes | Not Really | If Yes, Please Specify What Actions Will You Take to Preserve Your Environment? |
|---|---|---|---|---|
| | 9 students. | 22 students. | 2 students. | -Switch off electrical appliances; -Avoid leaving the water running for no reason; -Throwing waste in the garbage containers; -Collecting trash from the street and gardens. |
| What actions will you take (in the future) to preserve the environment in your neighborhood? | -Waste selective sorting. -Save water. -Save energy. -Nothing. | 16 students. 14 students. 23 students. 1 student. | (48.5%) (42.4%) (69.7%) (3%) | |
| If you have to choose one of the following school activities that will be noted in your evaluation, which one would you choose? | -Music -Painting -Environmental Education | 6 students. 8 students. 21 students. | (17.1%) (22.9%) (60%) | |
| Did you know that your school has an environmental club? | **Yes** | **No** | **If so, are you currently registered?** | |
| | | | **Yes** | **No** |
| | 28.1% | 71.9% | 12.5% | 87.5% |

To bridge the intention–action gap and support students' environmental engagement, schools play a crucial role in providing the necessary resources and support. Research has emphasized the significance of school environments in fostering pro-environmental attitudes and behaviors among students [42]. By offering relevant educational programs, creating opportunities for hands-on experiences, and establishing supportive structures, schools can empower students to translate their intentions into concrete environmental initiatives. The results of this study can provide valuable insights for policymakers and educators to develop effective strategies and programs to promote environmental education and awareness among high school students in Morocco.

3.2.2. Wastewater Treatment and Recycling Workshop

The present study conducted a workshop on wastewater treatment and reuse in a public high school in Morocco. Initially, the workshop was open to registered students who had completed the questionnaire. However, the participant number increased to 44 due to the request of some interested students. The objective of the workshop was introduced through an open question and answer session, which facilitated an environment for exchange among the participants [43]. The workshop was organized in the presence of teachers who were responsible for the school's environmental club. Two techniques of wastewater treatment were introduced, using a prototype of the planted filter and the MBBR system (moving bed bioreactor). The prototype was built in the laboratory of the chemistry of the faculty of sciences of Casablanca as part of doctoral research to simulate a system combining the two methods of treatment [44]. The use of this prototype as experimental material in a research laboratory stimulated the interest of both teachers and students.

After explaining the principles and functioning of each treatment method, the students were provided with the opportunity to manipulate and run the system. Water samples were taken before and after treatment to measure in situ parameters that determine water quality. With the help of a multi-parameter, the students could take pH measurements

and refer to their chemistry class to gain an understanding of the application of chemistry lessons in real life. Other parameters, such as conductivity and temperature, were also measured. In the end, the students participated in irrigating some trees near the school with the treated water that met the standards of reuse in Morocco. The workshop was successful in providing practical experience to the students and highlighting the importance of wastewater treatment and reuse in promoting sustainable development in the region.

3.2.3. The Output of the Mapping Workshop

The present study aims to discuss the outcomes of a mapping workshop conducted to design an educational garden in a school setting. The workshop was organized to engage students in the process of designing and planning a garden that would meet their needs and expectations. The workshop was held in a school setting and was attended by the registered students. The students were divided into three groups, each group accompanied by a teacher. The workshop started with an explanation of its purpose, and the students were informed that they would be designing an educational garden that would meet their needs and expectations. The students were then given 60 min to come up with their design ideas, and at the end of the session, each group presented their design ideas in 10 min. The groups were given the option to either stay in the room or go out and build their idea.

At the end of the mapping workshop, three drawings were submitted, as shown in Figure 9, which suggested planting trees around the sports field. The students expressed the need to have trees near the sports field, explaining that during their basketball and shuttlecock sessions, they were waiting for their turn in the sun and that having trees would allow them to stay protected from the sun. The drawings were submitted to the stakeholders, who were responsible for providing the necessary materials to realize the garden.

Community mapping has been recognized as a participatory approach that empowers individuals and communities to actively contribute to the design and planning processes [45,46]. By involving students in the mapping workshop, they were able to actively participate in identifying and documenting the features, resources, and potential opportunities within the garden space.

The positive outcomes of community mapping workshops have been reported in diverse settings. For instance, Caquard et al. [45] found that community mapping facilitated collaborative decision-making and improved spatial understanding among participants. Similarly, Elwood and Leszczynski [46] demonstrated that community mapping enhanced spatial cognition and contributed to place-based learning experiences.

Moreover, community mapping aligns with the principles of place-based education, where learning is connected to local environments and communities [47,48]. It promotes a deeper understanding of the local context, fosters a sense of connection to the environment, and enhances students' ecological literacy.

By providing students with an opportunity to actively contribute to the garden design through the mapping workshop, our study highlights the effectiveness of this approach in engaging students and incorporating their perspectives. It allows them to have a sense of agency and ownership in shaping their learning environment.

The present study showed that a mapping workshop can be an effective tool to engage students in the process of designing and planning a garden that meets their needs and expectations. The workshop resulted in the submission of three drawings, which were considered by the stakeholders for the realization of the garden. This approach to garden design allows for the participation of all stakeholders, including the students and teachers of the school, which could lead to the creation of a garden that is functional, educational, and sustainable. In conclusion, the mapping workshop is a promising approach to engage students in the process of designing an educational garden that meets their needs and expectations.

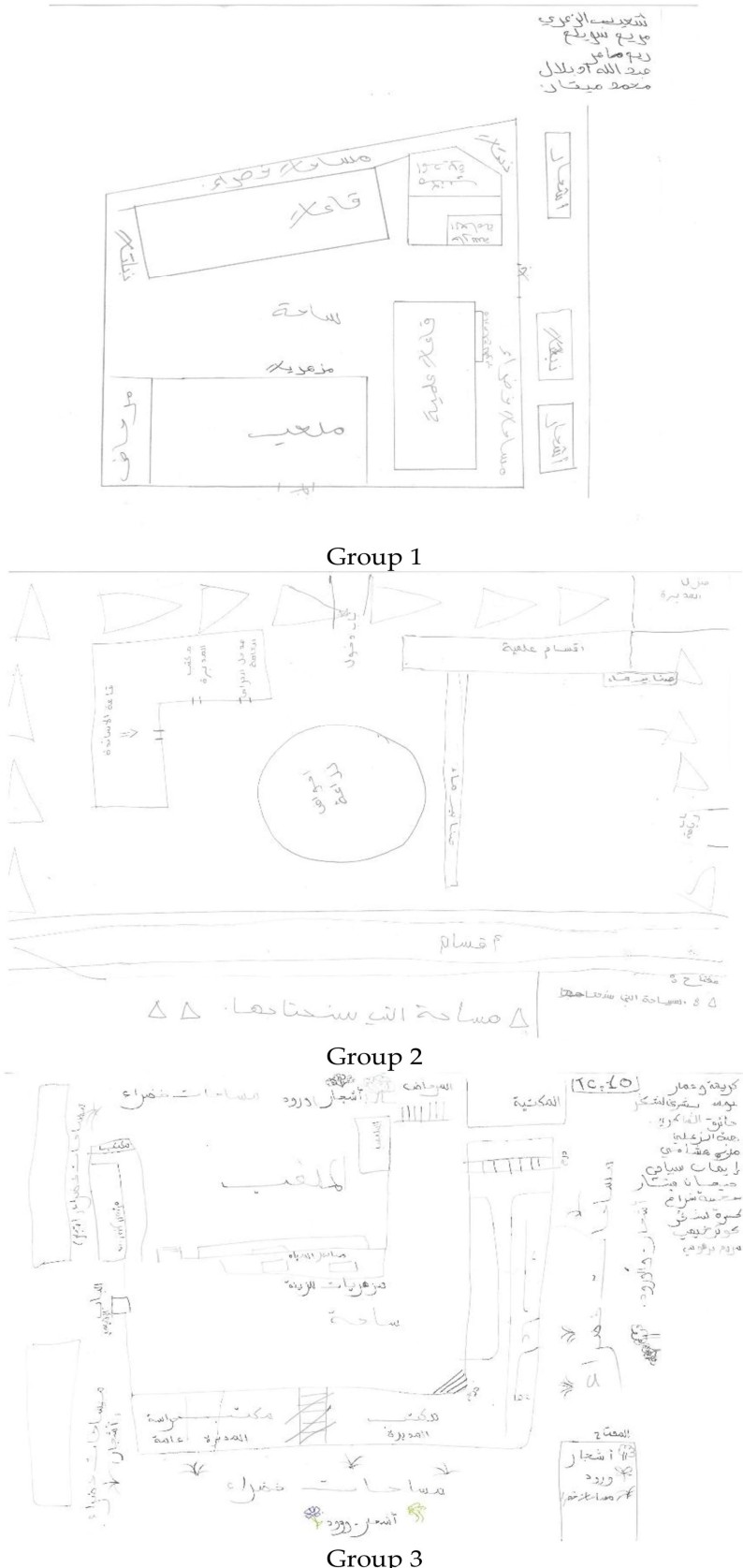

Group 1

Group 2

Group 3

**Figure 9.** Drawings made by the three groups of students for the educational garden in the school.

## 4. Conclusions

In conclusion, the emerging trend of incorporating a participatory approach to environmental education (EE) in schools reflects the recognition of the critical role of future generations in preserving and sustaining water resources [43]. This approach offers the potential to achieve outcomes that are unattainable through the conventional bureaucratic structure of educational institutions [49]. This article has explored how urban water management projects have been leveraged as a means of integrating environmental education into water engineering and chemistry initiatives, thereby creating an environmental education tool.

To begin with, this study has provided an overview of how such projects have served as a learning platform for various social categories, ranging from school and university students to individuals. The outcomes and benefits of this overview demonstrate how the content can be organized and successfully delivered in various formats. Secondly, a case study of a public school in Casablanca has been presented, which involves the implementation of a decentralized treatment and reuse project. The integration of stakeholders in a participatory approach has been essential in developing an interactive tool that bridges the gap between two distinct fields of study. The institutional differences inherent in this approach have presented several challenges; however, the awareness of the importance of future generations in sustainable water resource management has piqued the interest of several parties.

Thirdly, the collection of data to integrate a participatory approach to environmental education within a public school in Morocco may present several institutional challenges. The bureaucratic processes in public schools are often complex and may require navigating through multiple levels of administration, which may pose a significant obstacle to the effective implementation of such an approach. It may be necessary to identify the key stakeholders involved in the decision-making process and engage them in a constructive dialogue to ensure their buy-in and support for the project.

Paolo Freire, a renowned critical educator, emphasized the need for transformative pedagogy that challenges oppressive systems and promotes critical consciousness [50]. However, within the specific context of public schools in Morocco, these institutional challenges may pose hurdles to the successful application of Freirean principles.

Research conducted in Morocco highlights the bureaucratic hurdles and administrative complexities encountered within public schools. El Hamdi et al. [51] shed light on the bureaucratic challenges faced by teachers affecting the teaching and learning processes. Moreover, Benseddik [52] explored the institutional constraints and bureaucratic barriers within the Moroccan public education system, emphasizing the necessity for systemic changes to facilitate more effective and student-centered practices.

Within the Freirean framework, overcoming these challenges involves critical reflection, dialogue, and transformative action [50]. Engaging in critical dialogue with educational stakeholders, advocating for policy changes, and fostering a supportive environment that values critical thinking and student empowerment are crucial steps.

Moreover, the participatory approach may face resistance from traditional top-down management structures that are prevalent in many public schools. Convincing school administrators to adopt a more decentralized approach that empowers students, teachers, and other stakeholders may require a paradigm shift in their mindset, which may be challenging. Additionally, the participatory approach may require additional resources, such as training for teachers and staff, to effectively engage students and other stakeholders.

Fourthly, the exchange of knowledge and ideas between the project team, students, teachers, and administrative staff may also present several institutional challenges. The traditional hierarchical structure of public schools may limit the ability of students and other stakeholders to contribute meaningfully to the decision-making process [53]. Moreover, the project team may face resistance from teachers and staff who are resistant to change or who may not have sufficient knowledge about environmental education.

The main focus of the project is to design and construct a decentralized wastewater treatment and reuse system that can efficiently irrigate the educational garden of the school. The participatory approach adopted by the team not only helped in identifying the requirements for the garden's irrigation but also highlighted additional environmental challenges faced by the school. The project's success underscores the significance of integrating wastewater management and educational garden design, which can provide an eco-friendly solution for managing water resources. This study can serve as an exemplary model for other public schools in Morocco to adopt a participatory approach toward environmental education, particularly emphasizing the importance of sustainable wastewater management for a healthier environment.

**Author Contributions:** Conceptualization, H.N. and A.M.; methodology, H.N.; software, H.N.; validation, A.M., M.B. and H.N.; formal analysis, H.N.; investigation, H.N.; resources, B.E.A.; data curation, H.N. and N.C.; writing—original draft preparation, H.N.; writing—review and editing, H.N. and A.M.; visualization, H.N.; supervision, A.M.; project administration, M.B.; funding acquisition, F.A. All authors have read and agreed to the published version of the manuscript.

**Funding:** The study was carried out as a part of the first author's doctoral research project at TU Berlin. The research was funded through global center of spatial methods for urban sustainability scholarship awarded to the first author.

**Institutional Review Board Statement:** The Moroccan Ministry of Education and Research's Casablanca regional delegation approved this study.

**Informed Consent Statement:** Informed consent was obtained from all subjects involved in the study.

**Data Availability Statement:** The dataset underpinning this research is accessible upon request to the designated corresponding author. The availability of the data is subject to restrictions in alignment with privacy and ethical frameworks. The data, which were derived from workshops involving senior high school students in select private schools within Morocco, are not disseminated publicly owing to the inherent constraints associated with safeguarding participant privacy and adhering to ethical standards. To obtain access to the dataset, interested parties may contact the corresponding author.

**Acknowledgments:** We thank the German Research Foundation and the Open Access Publication Fund of the Technical University of Berlin for their invaluable support.

**Conflicts of Interest:** The authors declare no conflict of interest.

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
