# Peer review of "Linking Urban Water Management, Wastewater Recycling, and Environmental Education: A Case Study on Engaging Youth in Sustainable Water Resource Management in a Public School in Casablanca City, Morocco"

_education, doi:10.3390/educsci13080824_

Round 1

Reviewer 1 Report

 ·        The title of the paper should be re-written in order to provide a clear idea of the topic of the paper. 

·        The distinction between section 1 and 2 is not clear. I don´t understand why the authors select only 3 cases to present examples of educative projects on water management. A deeper literature review would be necessary and more appropriate in section 2. Moreover, figures 1 & 3 have not referred the authorship or the source. The same with table 1. As this section is a description of previous projects, the source of data should be cleared, as well as a brief explanation of why and how these 3 projects have been selected.  

·        There are some statements in the text which are not consistent and prove a wide lack of knowledge on the topic. e.g., “Research on wastewater management has been developed without considering the possibility of producing a learning tool for children”. In Europe and, concretely, in Spain, there is a large experience on educative projects on water management since more than 20 years! Literature review in the topic is needed!

·        In section 3, the structure and the goals of the paper are not very clear. Sometimes the authors refer to the paper as a “study”, while other they refer to an “action-research educative project”. This should be cleared.

·        A methodology section is recommended to be introduced. The structure of the paper is not clear. The method is briefly presented, for example, the questionnaire, but the typical content regarding the method (examples of questions, description of the sample, etc.) is presented in the results. This is confusing and does not correspond with the normal structure of a journal paper.  

English quality is fine, although some errors need to be correct.  

Author Response

Thank you for your valuable comments. We apologize for the careless errors we made in the manuscript. Following your suggestions, we have taken your remarks into account in a new, revised manuscript. We hope you find these revisions and the improved text satisfactory.
For point-by-point responses to your comments, please see the attachment.

Reviewer 2 Report

This is a good manuscript about a good research project. I am suggesting revisions because there is unrealised potential in the discussion of the research and there are epistemological inconsistencies in the manuscript which I believe could be improved by a more thorough review and use of more recent and more critical educational research. The concept of sustainable development and the SDGs have been discussed widely in the literature as well as reviews of participatory action research, and there is much that could be discussed in the context of this research. This would allow the authors to create more meaningful discussion as well as suggest further areas of research.

I have highlighted and commented extensively within the attached copy of the manuscript.

The English is good in this article. I have suggested a few minor changes of word use which I think are related to the differences between Arabic, French and English languages as well as the different political structures. 

commune -- municipality

dryness -- drought

There are some formatting inconsistencies such as the use of capitals - I have highlighted words that have these problems.

Author Response

Thank you for your valuable comments. We apologize for the careless errors we made in the manuscript. Following your suggestions, we have taken your remarks into account in a new, revised manuscript. We hope you find these revisions and the improved text satisfactory.
For point-by-point responses to your comments, please see the attachment.
To make it easy, we've replied to the revision document you sent.
Checkmarks mean that we have made the corrections.
Please contact us if you need more information.

Round 2

Reviewer 2 Report

on page 15 of 24 in the revised manuscript, the word "drought" is incorrect within the sentence on line 171. The sentence should be "actions, most students thought that they would engage in waste-selective sorting..."

I suggested that "drought" be used to replace the word "dryness" in figure 7

the quality is fine - only the incorrect use of the word "drought"

Author Response

Thank you for your careful review and valuable feedback. We appreciate your attention to detail and have duly noted the error on page 15, line 171 of the revised manuscript. The correct sentence should indeed be "actions, most students thought that they would engage in waste-selective sorting..." We will make the necessary corrections to ensure accuracy and clarity in our work.

Regarding your suggestion to use "drought" instead of "dryness" in Figure 7, we have carefully considered your input. While "drought" is a relevant term in the context of water scarcity and environmental concerns, we believe that "dryness" better conveys the specific condition being referred to in the figure. As such, we have decided to retain the term "dryness" in Figure 7.

Once again, we sincerely appreciate your efforts in reviewing our manuscript and providing valuable insights to enhance the quality of our research. If you have any further comments or suggestions, please do not hesitate to let us know.